# Loss of Neural Automaticity Contributes to Slower Walking in COPD Patients

**DOI:** 10.3390/cells11101606

**Published:** 2022-05-11

**Authors:** S. Ahmed Hassan, Leandro Viçosa Bonetti, Karina Tamy Kasawara, Matthew B. Stanbrook, Dmitry Rozenberg, W. Darlene Reid

**Affiliations:** 1Physical Therapy, Temerty Faculty of Medicine, University of Toronto, Ontario, ON M5G 1V7, Canada; karina.kasawara@utoronto.ca (K.T.K.); darlene.reid@utoronto.ca (W.D.R.); 2Institute of Health Policy, Management and Evaluation, University of Toronto, Ontario, ON M5T 3M6, Canada; 3Department of Physical Therapy, Universidade de Caxias do Sul, Caxias do Sul 95070-560, Rio Grande do Sul, Brazil; leandrovbonetti@gmail.com; 4Post-Graduation Program in Health Sciences, Universidade de Caxias do Sul, Caxias do Sul 95070-560, Rio Grande do Sul, Brazil; 5Division of Respirology, Temerty Faculty of Medicine, University Health Network, Toronto, ON M5T 2S8, Canada; matthew.stanbrook@uhn.ca (M.B.S.); dmitry.rozenberg@uhn.ca (D.R.); 6Ajmera Transplant Program, Toronto General Hospital Research Institute, Toronto, ON M5G 2C4, Canada; 7KITE-Toronto Rehabilitation Institute, University Health Network, Toronto, ON M5G 2A2, Canada; 8Interdepartmental Division of Critical Care Medicine, University of Toronto, Toronto, ON M5B 1T8, Canada

**Keywords:** automaticity, pulmonary disease, chronic obstructive, prefrontal cortex, spectroscopy, near-infrared, gait

## Abstract

The physical impairments (e.g., slower walking speed) in patients with chronic obstructive pulmonary disease (COPD) have been attributed to peripheral characteristics (e.g., muscle atrophy). However, cognitive impairment may compromise motor control including walking automaticity. The objective of this study was to investigate PFC neural activity, evaluated using changes in oxygenated hemoglobin (ΔO_2_Hb), during preferred paced walking (PPW) in COPD patients and age-matched controls. The ΔO_2_Hb from the left and right dorsolateral PFC was measured using functional near-infrared spectroscopy. Fifteen COPD patients (age: 71 ± 8) and twenty age-matched controls (69 ± 7 years) participated. Two-way mixed ANOVA demonstrated that O_2_Hb in both groups decreased during PPW from the start (quintile 1; Q1) to the end (quintile 5; Q5) in the left dorsolateral and medial PFC. Q1 was comprised of the data during the first 20% of the task, while Q5 included data collected in the last 20% of the task duration. PPW duration ranged between 30.0 and 61.4 s in the control group and between 28.6 and 73.0 s in COPD patients. COPD patients demonstrated a higher O_2_Hb in Q5 compared to the negative O_2_Hb in controls in the right medial and dorsolateral PFC during PPW. PPW velocity was lower in COPD patients compared to controls (1.02 ± 0.22 vs. 1.22 ± 0.14 m/s, *p* = 0.005). Healthy older controls exhibited automaticity during walking unlike patients with COPD. The lesser decrease in O_2_Hb in COPD patients may be attributed to increased executive demands or affect-related cues (e.g., pain or dyspnea) during walking.

## 1. Introduction

Neurologic impairment can compromise motor control including the automaticity of walking [1], which could affect patients with chronic obstructive pulmonary disease (COPD) due to their well-documented cognitive impairment [2]. In COPD, impaired executive function (attention, strategizing) involving activation of the prefrontal cortex (PFC) has been attributed to the effects of cigarette smoking, hypoxemia, inflammation, atherosclerosis and aging on neuro-cortical tissue [2]. Over-activation of the PFC has been documented in COPD patients during dyspnea-related cues [3]; however, PFC activation during physical tasks is not well described.

Although COPD patients often walk slowly [4], have impaired balance and more falls [5], physical impairments have been primarily attributed to dyspnea and peripheral characteristics (i.e., muscle atrophy, altered muscle fiber characteristics) [6]. In addition, several physiological factors contribute to physical activity levels. Higher intensity of daily physical activity is positively associated with the central hemodynamic response of increased cardiac output, systemic vascular conductance (i.e., ease of blood flow at a given pressure difference) and systemic oxygen delivery in COPD patients [7]. Moreover, physically active COPD patients have better ventilatory capabilities and are able to expand tidal volume as well as maintain inspiratory reserve during walking [8]. Physical activity has been shown to improve cognitive function in COPD patients but the converse of whether altered PFC neural activity limits walking in COPD patients has rarely been studied [4].

Usual or preferred paced walking (PPW) is considered to be a relatively automatic physical task for healthy adults [1]. Automaticity minimizes attentional demands of executive control, but this can be impaired in various disorders [9]. Loss of automaticity due to central impairments can be compensated by increased neural activity indicative of greater executive demands [1]. Neural activity during walking can be evaluated by PFC oxygenated hemoglobin changes (PFC ΔO_2_Hb) using continuous-wave functional near infrared spectroscopy (fNIRS) [1,4]. fNIRS provides real-time data and allows evaluation during movement [10]. The primary aim of this study was to investigate PFC neural activity (ΔO_2_Hb) during PPW in COPD patients and age-matched controls. A secondary aim was to evaluate PFC ΔO_2_Hb during spelling backwards (SB). It was hypothesized that PFC O_2_Hb will decrease during PPW in controls (suggesting automaticity) unlike in COPD patients [10,11]. Moreover, PFC O_2_Hb was hypothesized to increase during SB in both groups because it requires ongoing attention and information processing.

## 2. Materials and Methods

This is a secondary analysis of a study that previously reported outcomes throughout the duration of the single (PPW) and dual tasks [4], whereas this report focuses on the PFC ΔO_2_Hb from the first (Q1) to the fifth (last) quintile (Q5) of single task durations (i.e., PPW and SB). Q1 was the average ΔO_2_Hb during the first 20% of the task’s duration, whereas Q5 was the average ΔO_2_Hb during the last 20% of the task duration. After screening and informed consent, an fNIRS device (FNIR100W-1, Biopac) was secured over the participant’s forehead to evaluate PFC ΔO_2_Hb during the performance of SB and PPW. Gait velocity was assessed by a pressure-sensitive Zeno Walkway. Participants walked back and forth in the center of the mat for 30 m which included making six passes back and forth. When participants reached the end of the 5 m mat, they got off and walked an additional 0.5 m to turn around before making the next pass. This was repeated until three back and forth rounds were completed [12,13]. PPW is the focus of this study as it is deemed to be more relevant than fast-paced walking because PPW is more commonly involved in activities of daily living. During SB, participants spelled five letter words backwards from a list of 100 unique words for one minute. Words were chosen carefully to avoid homonyms and prevent confusion when the researcher recited them to be spelled backwards. Methods are outlined in more detail in Hassan et al. [1,4].

Preprocessing of fNIRS data to attenuate physiologic artifacts (respiration and pulse) included the application of a low-pass, finite impulse response, filter with hamming order 57 and cut-off frequency of 0.05 [1,4]. Motion artifacts in the collected fNIRS data were identified as sharp peaks by visually inspecting graphs of collected data before any processing. Subsequently, the Sliding Motion Artifact Rejection filter was applied with default settings, developed by Ayaz et al. (2010), that uses an algorithm to identify and remove motion artifacts [14]. The algorithm scans the data to identify spikes or burst noises with values that are either much higher or lower than normal cortical activity using high and low thresholds of light intensity measured at 730 nm and 850 nm.

Using SPSS (V.28), a Fisher’s exact test examined group differences in the proportion of participants answering yes to “experience unreasonable breathlessness” on the American College of Sports Medicine screening questionnaire [4]. Mixed two-way ANOVAs examined differences between time (Q1 vs. Q5) and groups (COPD vs. controls) followed by post-hoc *t*-tests if group differences or group*time interaction were significant. Point biserial correlation was performed between the categorical variable (presence or absence of dyspnea) and continuous variable (ΔO_2_Hb during PPW from Q1 to Q5). Mean and standard deviation (SD) are provided unless otherwise stated.

## 3. Results

Fifteen COPD patients (forced expiratory volume in one second (FEV_1_: 52.3 ± 20.3%pred) and 20 age-matched controls (FEV_1_: 92.6 ± 11.7%pred) completed the study. The two groups had similar BMI, age (71 ± 8 vs. 69 ± 7 years, respectively) and sex ratios (9M:6F vs. 9M:11F, respectively) [4]. COPD patients compared to controls had 2.3-fold more comorbidities (*p* = 0.001) that showed frequencies of fractures (*n* = 7) and osteoporosis (*n* = 7) most commonly whereas heart failure and cardiovascular conditions were less common (*n* = 3). COPD patients took four times more medications (*p* = 0.001), had a 69.4% shorter duration of the single-leg-stance and had lower Montreal Cognitive Assessment scores (*p* = 0.032) [4]. Moreover, COPD patients had a 7.3-fold higher experience of unreasonable breathlessness (73% vs. 10%, *p* < 0.001) as evaluated by the American College of Sports Medicine questionnaire [15].

PPW duration ranged between 30.0 and 61.4 s in the control group and between 28.6 and 73.0 s in COPD patients. Using Q1 and Q5 data, two-way mixed ANOVA demonstrated that O_2_Hb in both groups decreased during PPW from Q1 to Q5 in the left dlPFC and mPFC (Figure 1) indicating automaticity. In contrast, COPD patients demonstrated a higher O_2_Hb in Q5 compared to the negative O_2_Hb in controls in the right mPFC and dlPFC during PPW (*p* = 0.015 and 0.026, respectively) (Figure 1); the Q1 to Q5 ΔO_2_Hb was less negative in COPD patients versus controls (right mPFC −0.118 vs. −1.040, *p* = 0.028; right dlPFC 0.127 vs. −0.575; *p* = 0.032). Furthermore, as expected due to high cognitive demands, O_2_Hb increased during SB from Q1 to Q5 in four PFC regions (left and right medial (mPFC) and dorsolateral (dlPFC)) (Figure 1). Lastly, PPW velocities were lower in COPD compared to controls (1.02 ± 0.22 vs. 1.22 ± 0.14 m/s, *p* = 0.005).

Moreover, a point-biserial correlation was run to determine the relationship between dyspnea and ΔO_2_Hb from Q1 to Q5 in the four PFC regions, because a negative difference would be indicative of decreased PFC activity at the end of the task. The results indicated non-significant correlations (*p* > 0.05) between dyspnea and PPW ΔO_2_Hb from Q1 to Q5 in right medial PFC (rpb = −0.220, 95% CI: −0.533 to 0.146), right dlPFC (rpb = −0.283, 95% CI: −0.579 to 0.080), left medial PFC (rpb = −0.116, 95% CI: −0.452 to 0.248), and left dlPFC (rpb = −0.077, 95% CI: −0.420 to 0.285). Negative correlations are indicative of decreased PFC activity (from Q1 to Q5) in those who did report dyspnea.

## 4. Discussion

Healthy older controls exhibited automaticity during walking as indicated by decreases in O_2_Hb, a proxy for neural activity in four regions of the PFC. This reflects the ability to control steady state walking while minimizing executive control demands [1]. In contrast, COPD patients did not decrease O_2_Hb in the right mPFC and dlPFC and tended to have a smaller decrease in the left mPFC (*p* = 0.065) during walking, which may be attributed to increased executive demands or affect-related cues during walking. Furthermore, as expected, automaticity was not observed during the SB task and served to validate the data collected for the walking task in both groups.

Neural activity may be affected by several factors such as comorbidities. The most common comorbidities in COPD patients in this study were fractures and osteoporosis; however, literature shows limited evidence of their impact on changes in neural activity in the frontal cortex. Although decreased blood velocity of the middle cerebral artery has been reported after humeral fractures [16], none of the participants in this study reported upper limb fractures. Anxiety [17], diabetes [18] and cardiovascular conditions were reported less commonly in this study (*n* = 3), which are associated with decreased blood flow and oxygenation to the frontal cortical region [19,20]. Given the low prevalence of these comorbidities, their contributing relationships with findings can only be speculated.

The lower decrease in neural activity (ΔO_2_Hb) during PPW in COPD patients, in contrast to age-matched controls, could be due to greater dyspnea and pain, commonly experienced by these patients. Neuronal activity in the mPFC has been documented during dyspnea-related cue words in COPD patients and fear-related memory of chronic pain [3]. Neural activity related to the anticipation of dyspnea can subsequently dictate behaviour [21]. Consequently, fear of dyspnea and increased right mPFC activation in COPD patients may lead to self-limitation of physical activity [22] including slower walking velocity. However, biserial correlation analysis did not suggest a significant correlation between dyspnea and ΔO_2_Hb during PPW. In contrast, the greater decreases in PFC O_2_Hb during walking in age-matched controls may, in part, be reflective of a lower symptom burden. A numerical dyspnea scale or biserial correlation analysis of a larger sample may provide stronger, significant correlations.

Neural demands in the dlPFC increase during activities that require attention and working memory [1,23]. The sustained right dlPFC O_2_Hb in COPD patients, in contrast to age-matched controls, may reflect continued attention to prioritize posture and minimize fall risk even during the low demands of PPW on an indoor flat surface. Together with previously described balance deficiencies [5] and limb muscle dysfunction [6] noted in COPD, cognitive impairment may further contribute to slow walking velocity. Reduction of gait speed is a well-described “posture first” compensatory strategy to maintain balance in older adults and other conditions [24] and may serve to prevent falls in COPD patients. Unfortunately, a slower walking speed reduces the ability to do some daily activities including the required gait velocity to safely cross the street at a traffic light commonly set at 1.2 m/s [25]. Slower gait speed is also associated with a greater risk of falls [26].

Cognitive impairments in COPD patients, compared to older adults, may not only compromise dual tasking [4] but also limit an apparently simple physical task such as walking. The patterns of PFC neural activity provide strong evidence of impaired automaticity during walking in COPD patients compared to age-matched controls. However, automaticity can improve from single and dual task training as shown in healthy adults and disorders such as stroke and amputees [9]. Moreover, there appears to be a transfer effect from cognitive training to physical tasks, such as the timed-up-and-go, in older adults [27]. Further, physical and cognitive training appears to induce distinct changes related to reaction time in dual task performance [28]. Thus, these training strategies, previously applied to other populations could potentially improve the automaticity of walking in COPD patients. A particular strength of cognitive therapy is that it may be more safely and readily applied when access to face-to-face physical training is constrained (i.e., COVID-19 pandemic or travel distance to rehabilitation). In the event that reversibility of cognitive impairment is limited, simplification of tasks will be warranted (i.e., avoid cognitive demands during walking).

This study is limited by a convenience sample of age-matched controls and COPD patients that may not be generalizable to those with greater cognitive impairments. Secondly, relative changes rather than absolute measures of O_2_Hb from continuous-wave fNIRs may limit interpretation. Thus, it is not possible to examine the starting level but only the change in O_2_Hb. In contrast, frequency domain and time domain NIRS enable absolute measures but severely limit movement due to their design. Notably, continuous wave portable fNIRS device can acquire real-time data during movement such as walking. Lastly, deoxygenated hemoglobin (HHb) data is not reported, which might be beneficial in providing a more in-depth understanding of cortical activation. HHb is a marker of oxygen extraction and provides a measure of total blood volume delivered when combined with O_2_Hb [29,30].

In conclusion, although older healthy adults demonstrated neural efficiency during walking, COPD patients did not. This lack of automaticity may be an important contributing factor to walking impairment. Slower walking, poor balance and more falls have been primarily attributed to dyspnea and muscle characteristics. Moreover, central hemodynamic impairments [7] and limitations in tidal volume and inspiratory reserve are associated with limitations of physical activity in COPD patients [8]. This report provides evidence of how neural inefficiency may contribute to slower walking in COPD patients even when ambulating on a flat surface indoors.

Future studies may utilize fNIRS devices that allow more comprehensive coverage of other brain regions to allow measurements of neuronal activity. This may be important as other areas may be activated during motor tasks such as the premotor and primary motor cortex, caudal supplementary motor area and posterior occipital temporal areas [31,32]. Moreover, studies with a larger sample size should be conducted and interpreted along with deoxygenated hemoglobin data to confirm and expand the observed findings. Further evaluation of automaticity in COPD patients may allow a greater understanding of mechanisms underlying activities of daily living and functional limitations in larger cohort studies.

## Figures and Tables

**Figure 1 cells-11-01606-f001:**
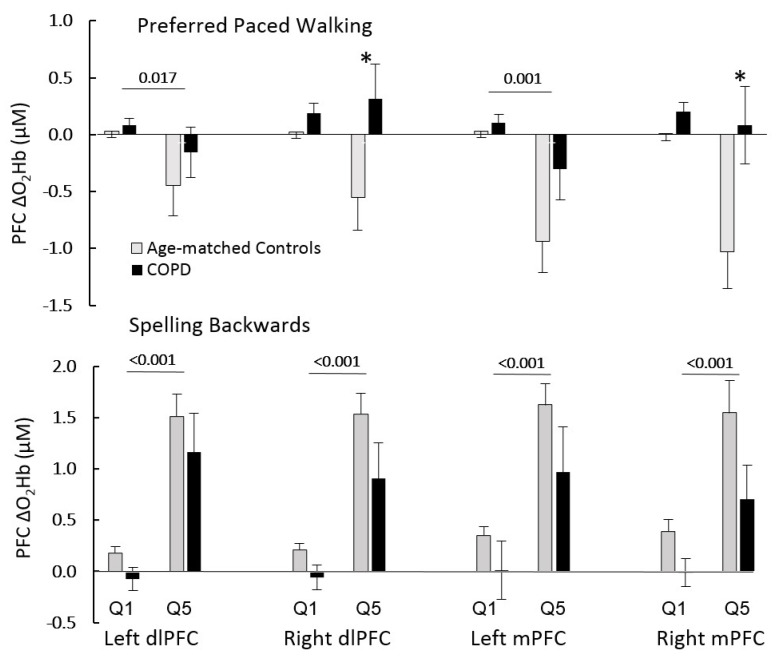
Changes in medial (mPFC) and dorsolateral prefrontal cortex (dlPFC) O_2_Hb from the first (Q1) to the last quintile (Q5) of the single task durations: preferred paced walking (**upper panel**) and spelling backwards for 60 s (**lower panel**). Means and SE are shown. Horizontal bars indicate the main effects between Q1 and Q5 (mixed 2-way ANOVA). * COPD different than age-matched controls at *p* < 0.05.

## Data Availability

Available upon request.

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
