# Peer review of "Loss of Neural Automaticity Contributes to Slower Walking in COPD Patients"

_cells, 2022, doi:10.3390/cells11101606_

Round 1

Reviewer 1 Report

This is an interesting study by a group that has a lot of experience in this topic of interest. The study provides novel insights into the role of prefrontal cerebral oxygenation during daily living tasks in patients with COPD. Please see my comments below. 

  1. The introduction section is well written. However, the study's aim and hypothesis are not clearly defined. Please revise accordingly.
  2. Please address in the introduction and discuss in the discussion sections, the studies by the group of Vogiatzis et al; that investigated the physiological determinants (respiratory mechanics and central and lower muscle oxygen availability) of walking intensity in patients with COPD; DOI: 10.1152/japplphysiol.00301.2014 DOI: 10.1152/japplphysiol.00379.2013
  3. In the methods section, it is important for the clarity of the manuscript, the authors clearly define the variables previously published in the study by Hassan SA et al, 2020 and represent them in this study. 
  4. The results section is clear and concise.
  5. Discussion section: Lines 109-111. This is a very interesting interpretation of the main study findings. Authors might need to perform a correlation analysis between dyspnea and prefrontal cortex oxygenation to support this argument. 
  6. The authors need to further discuss the impact of comorbidities in the major findings of the study (ie., their impact on patients' capacity to deliver sufficient oxygen to the brain)
  7. Study limitations. Authors report that ^Secondly, relative changes rather than absolute measures of O2Hb from continuous-wave fNIRs may limit interpretation^. Why does this constitute a study limitation? Please provide arguments.
  8. A section outlining future research considerations based on the findings of the study is necessary. 

Reviewer 2 Report

This communication article aims to compare neural activity changes at the PFC during preferred paced walking in COPD patients and age-matched controls. It also aimed to determine whether ΔO2Hb decreases in both groups. This communication article is well structured and well written and with an excellent Discussion section. However, some minor points need to be corrected/clarified.

  • Abstract 

Lines 24-26: Please rephrase the objective of this study. In particular, I would remove “decreases (an indicator of automaticity); line 25”. If this is a fact, the novelty of your work will be questioned. As a reader, when I read this part of your manuscript for the first time, I thought that ΔO2Hb always decreases during PPW (not necessarily true; not all subjects/patients react the same), and the authors wanted to check this phenomenon on their patients as well as their control group. You can use more general sentences. My suggestion: “The objective of this study was to investigate PFC neural activity changes; evaluated using changes in oxygenated hemoglobin (ΔO2Hb); during preferred paced walking …”

Lines 29-30: A well-written abstract should be reflective of the overall contents of the research article, and the reader should be able to grasp the subject without referring to the main text. I recommend adding more information about Q1 and Q5 in the abstract. How long did the whole walking task last? How long did each quintile last? Concerning this point that the abstract should be self-explanatory, it is unclear what Q1 and Q5 are. The authors could change the sentence to “… during PPW from the start (quintile 1; Q1) to the end (quintile 5; Q5) in the left …”

  • Introduction

Line 60: In line with the first comment, please rephrase this sentence. If the authors would like to keep it, at least 2-3 references (Preferably not self-citation) should be provided to support that automaticity leads to a decrease in ΔO2Hb; also, please indicate that this is a TYPICAL hemodynamic response pattern during PPW.

  • Materials and Methods

Line 68: More information about the walking task protocol is required. How long did each quintile last? 6 min? Statistical analysis should be a bit explained in detail. Please also specify the methods used to process motion artifacts. This should also be noted since fNIRS is sensitive to artifacts (head/body movements and physiological noise) and preprocessing for fNIRS data is an important step.  

  • Results

Lines 79-80: What does FEV stand for? Forced expiratory volume? What about FEV for age-matched controls? How many females and males were in each group?

  • Discussion

To better understand and interpret neural activity in fNIRS studies, it is highly recommended to assess and report both O2HB and deoxygenated hemoglobin (HHb) [1]. Single measures (O2HB or HHb) may not be sufficient to characterize the neurovascular response of neuronal tissue [2]. Please also indicate this point in the limitation paragraph of the Discussion section.

[1] Herold, Fabian, et al. "Applications of functional near-infrared spectroscopy (fNIRS) neuroimaging in exercise–cognition science: a systematic, methodology-focused review." Journal of clinical medicine 7.12 (2018): 466

[2] Tam, Nicoladie D., and George Zouridakis. "Temporal decoupling of oxy-and deoxy-hemoglobin hemodynamic responses detected by functional near-infrared spectroscopy (fNIRS)." Journal of Biomedical Engineering and Medical Imaging 1.2 (2014).
